

# Sleep disorder and apnea events detection framework with high performance using two-tier learning model design

Recep Sinan Arslan

Computer Engineering, Kayseri University, Kayseri, Turkey

## ABSTRACT

Sleep apnea is defined as a breathing disorder that affects sleep. Early detection of sleep apnea helps doctors to take intervention for patients to prevent sleep apnea. Manually making this determination is a time-consuming and subjectivity problem. Therefore, many different methods based on polysomnography (PSG) have been proposed and applied to detect this disorder. In this study, a unique two-layer method is proposed, in which there are four different deep learning models in the deep neural network (DNN), gated recurrent unit (GRU), recurrent neural network (RNN), RNN-based-long term short term memory (LSTM) architecture in the first layer, and a machine learning-based meta-learner (decision-layer) in the second layer. The strategy of making a preliminary decision in the first layer and verifying/ correcting the results in the second layer is adopted. In the training of this architecture, a vector consisting of 23 features consisting of snore, oxygen saturation, arousal and sleep score data is used together with PSG data. A dataset consisting of 50 patients, both children and adults, is prepared. A number of pre-processing and under-sampling applications have been made to eliminate the problem of unbalanced classes. Proposed method has an accuracy of 95.74% and 99.4% in accuracy of apnea detection (apnea, hypopnea and normal) and apnea types detection (central, mixed and obstructive), respectively. Experimental results demonstrate that patient-independent consistent results can be produced with high accuracy. This robust model can be considered as a system that will help in the decisions of sleep clinics where it is expected to detect sleep disorders in detail with high performance.

## INTRODUCTION

Sleep apnea can be considered as sleep disorder that disturbs person's sleep. There are three types of sleep apnea as obstructive, central, and mixed. Obstructive sleep apnea occurs due to improper functioning of the upper respiratory tract. Central sleep apnea occurs when the brain fails to generate signals to control breathing muscles. Finally mixed sleep apnea which occurs due to central apnea persisting even after obstructive sleep apnea (OSA) disappeared with positive air pressure therapy (*Javaheri et al., 2017*; *Lee & Sundar, 2021*; *Selim & Ramar, 2020*).

Corresponding author
Recep Sinan Arslan,
sinanarslanemail@gmail.com

Obstructive sleep apnea is one of the most important sleep disorder syndromes seen in the respiratory tract (*Yeghiazarians et al., 2021*). This syndrome causes snoring and respiratory effort to overcome the resistance that occurs in the upper respiratory tract (*Benjafield et al., 2019*). It shows that an average of 1 million people worldwide is affected by this disease (*Drager et al., 2017*). OSA syndrome can cause several other diseases such as heart disease, and early diagnosis and treatment are therefore very important (*Floras, 2018*; *Linz et al., 2018*). The standard approach in the detection of this syndrome is Polysomnography (PSG) (*Patil et al., 2019*). A series of sensors such as ECG, EEG, SpO2, respiratory effort and airflow are connected to the patient and recorded by PSG during a night's sleep (7:30-8:00 h on average). Thus, it is possible to analyze these data later and to detect diseases (*Faber, Faber & Faber, 2019*). Hypopnea is defined as a type of respiration in which air flow is reduced by at least 50% and does not prevent air entry into the body (*Hamnvik, 2021*). In obstructive apnea, while breathing is completely obstructed, there is a serious decrease instead of obstruction in hypopnea. This is the main difference between obstructive sleep apnea and hypopnea. Since every patient data composed from around 700 epochs (30s each), analyzing sleep, and calculating the apnea hypopnea index (AHI) is time consuming process that must be done by a sleep doctor or sleep expert (*Mendonca et al., 2019*). The apnea- hypopnea index can be calculated by 60 × (apneas + hypopneas)/ total sleep-in minutes.

Obstructive sleep apnea syndrome (OSAS) severity detection could be important for developing new methods. For example, revealing prediction between man and woman (*Schutte-Rodin et al., 2021*), neck circumference and OSAS relation (*Bixler et al., 2001*). In 1970s, healthy subjects classified as event rate <5 apneas per hour and it is accepted as a standard for defining disease and no disease (*Ahbab et al., 2013*). Hypopnea was defined as decreased respiratory events due to a decrease in oxygen saturation or arousal. However, a sense that this decline is physiologically significant is to be expected (*Heise, Yi & Despins, 2021*; *Malhotra et al., 2021*). Initially OSA was defined by 30 apneas over the night, then it is defined by an index which is defined by the number of apneas/hours to diagnose the disease. OSAS severity can be assessed as mild when it is occurred between 5–15, moderate if AHI index between 15 and 30, and severe when AHI >30 (*Kumar et al., 2014*; *Taranto-Montemurro et al., 2019*; *Gauld et al., 2022*).

Electrocardiogram (ECG) is one of the most reliable physiological signals containing information about central cardiovascular function, respiration, and electrical activity of heart. For this reason, we see several studies on AHI index screening by using machine learning and deep learning methods as in *Banluesombatkul, Rakthanmanon & Wilaiprasitporn (2018)*, *Ivanko, Ivanushkina & Rykhalska (2020)*, *Li et al. (2018)*, *Erdenebayar et al. (2019)*. Many patients with OSA report a history of shortness of breath associated with hypopnea. Therefore, oxygen level decreases from 90% to lower levels which can be considered as hazardous and requires immediate medial actions. So that, single channel blood oxygen saturation (SpO2) is also used to predict AHI index by using machine learning or deep learning-based approaches as in the *Gutierrez-Tobal et al. (2019)*, *Álvarez et al. (2020)*, *Barroso-García et al. (2021)*, electroencephalographic (EEG) signals require more than one probe to sense individual's effort for breathing and

detection of OSA and its type. This signal is also being used for AHI index calculation as in the *Wang et al. (2021)*.

Computer engineers work with sleep doctors for sleep stage scoring and detection of sleep apnea as well as AHI index calculation effectively over the past decade (*Hassan & Haque, 2017*; *Supratak et al., 2017*; *Hilmisson, Lange & Duntley, 2018*). To do so, some researchers employed machine learning techniques such as the hidden Markov model for feature extraction and classification (*Song et al., 2016*). Machine learning is generally used in the structure of supervised learning to detect sleep apnea (*Ramachandran & Karuppiah, 2021*). In addition, it is possible to prefer unsupervised or reinforced learning in cases where expert support cannot be provided. Among the machine learning models, while CNN networks are used in image-based models, different model structures such as DNN, RNN, LSTM, GRU can be preferred. The choice between these models determines the ability of the data to recall the past, and this has a significant impact on the detection of sleep disorders (*Li et al., 2018*). Due to increasing number of methods, deep learning-based models are being employed in OSA detection (*Yao et al., 2020*; *Wang et al., 2019*). Although deep learning-based solutions for OSA detection was improved, there are still limitations as employing single channel for detection and acquiring limited accuracy levels as well as using publicly available datasets (*Shen et al., 2021*).

In this study, a two-layer classifier was designed in which DNN, LSTM, RNN and GRU networks are used in the first layer, and a pre-trained model is used in the second layer, which is tested with 11 different ML algorithms. Thus, it is aimed to find solutions to the constraints of deep learning networks and to achieve high performance. The conditions of the patients are first classified as Apnea-Hypopnea-Normal, and in the second stage, the apnea types are determined as Mixed, Central and Obstructive. In this way, it was ensured that the disease status was determined more clearly and in detail.

Compared to previous studies, the following improvements were made in this study:

- A unique multiclass model which detects apnea types in two layers occupied from LSTM, GRU, DNN and RNN networks in the first layer and pre-trained decision-making ML model placed in the second layers is proposed.
- Unlike similar studies, after Apnea-Hypopnea-Normal status is determined, the events of apnea types are detected as mixed, central, or obstructive with high performance in the second stage. Thus, six different types of classification of the disease were made.
- It was studied with a discrete signal instead of a continuous signal.
- By using 23 different features, more distinctive features of the patient were captured. While an accuracy value of 95.76% was obtained, an increase of 1.19% was achieved by using the two-layer structure compared to deep learning models.

This study has a comprehensive structure and is divided into sections as follows. In "Related Works", studies on the detection of sleep apnea were examined and analyzed in separate sections on a method-based basis. In "Methodology", the two-tier architecture proposed in this study is explained in detail. In addition, the stages of model creation, dataset collection, deep learning and machine learning model structures are explained in

this section. In "Obtained Results ", the test results of the proposed model are given. In "Discussion", the proposed model is compared with similar studies and information about some of its limitations and unsolvable situations of the problem is given. In the light of this information, future studies are mentioned. In the last part, a general evaluation of the study was made. After the references, it was shared with the publication in two separate annexes.

## RELATED WORKS

To present the current work' differences from similar deep learning based and machine learning based studies, this section is prepared and obtained results briefly given in Table 1.

### Deep learning based studies

One of the most common methods for detecting sleep disorders is the convolution neural network (CNN), LSTM, RNN and Bi-LSTM structures in deep learning structure. It has been seen that in CNN-based networks, it is mostly based on the processing of images of EEG signals, and in LSTM, RNN and Bi-LSTM networks, analysis is made over numerical data, as in our study. Studies are concerned with detecting sleep disorders as classification problems and calculating the AHI index. In this section, examples and current studies using deep learning structure are evaluated.

*Wang et al. (2019)* detected sleep apnea with 87.6% accuracy with a CNN network called LeNet-5. The model trains single-channel ECG data on a one-dimensional CNN network and is based on the approach of evaluating adjacent segments. *Shen et al. (2021)*, like other studies, proposed a classification method based on 1-D CNN network and time-dependent weight loss. Their accuracy was 89.4% for noninvasive wearable devices. *Urtnasan et al. (2020)*, proposed a CNN-based model using segmented SMC and apnea dataset to determine the severity of sleep apnea. They successfully differentiate the mild and severe apnea with an accuracy of 99% by using segmented dataset which employs ECG signal (*Urtnasan et al., 2020*). Another ECG based study conducted by *Hedman et al. (2021)*, proposed an LSTM-based network for classifying 35 labeled patients by looking at the long-term dependencies of the ECG signals they received as a single channel over the PyhsioNet-ECG dataset. They achieved 97.1% accuracy in detection of sleep apnea events. *Iwasaki et al. (2021)*, proposed a SAS method and LSTM to analyze R waves on ECG records to predict moderate to severe SAS. They obtained 100% accuracy in differentiation of moderate and severe apnea by using their own dataset (*Iwasaki et al., 2021*). They used PhysioNet Apnea-ECG dataset occupied from 70 patients and obtained 91.7% accuracy diagnose level with using different forms of RNN and single channel ECG signal. *De Falco et al. (2019)* proposed a deep neural network-based model for the detection of sleep apnea using ECG signals. Their accuracy was around 73% on Sleep Heart Health Study database. *Chang et al. (2020)*, ECG signals were used as a single channel in training a 1D D-CNN network for the detection of sleep apnea, their model obtained 87.9% accuracy. *Sheta et al. (2021)*, worked on the reduction of noise *via* filter, extracting features from the ECG signal and developing thirteen machine learning and four deep learning algorithms on ECG signals and the automatic classification system called CAD in the detection of sleep apnea. Their model achieved an accuracy of 86.25 on Physionet's CinC challenge-2000 database.

**Table 1 Brief comparison of similar works with presented work.**

| Reference | Method | Dataset | Channels | Accuracy |
|---|---|---|---|---|
| Erdenebayar et al. (2019) | 1D CNN and 2D CNN, RNN | Samsung Medical Center | ECG | 99% |
| Wang et al. (2019) | LeNet-5 | PhysioNet Apnea-ECG | Single channel ECG | 87.6% |
| Shen et al. (2021) | Deep learning CNN+ weighted-loss time-dependent classification | Apnea-ECG | ECG | 89.4% |
| Urtnasan et al. (2020) | CNN | SMC and sleep apnea dataset | ECG | 99% (binary classifier for mild and severe |
| Hedman et al. (2021) | LSTM | PhysioNet Apnea-ECG | ECG | 97.1% |
| Iwasaki et al. (2021) | LSTM | Own dataset | ECG signals | 100% (moderate and severe apnea) |
| De Falco et al. (2019) | Deep learning, Grid Search | Sleep Heart Health Study | ECG | 72.91% |
| Chang et al. (2020) | Deep learning, CNN | MIT PhysioNet Apnea-ECG | Single channel ECG | 87.9% |
| Sheta et al. (2021) | Deep learning and machine learning, CNN+LSTM | Physionet's CinC challenge-2000 | ECG signals | 86.2% |
| Yang et al. (2022) | Deep learning, multi-model fusion | Apnea-ECG | ECG | 90.3% |
| Wang et al. (2022) | Deep learning, LSTM | Own dataset | EEG channels | 92.73% |
| Sharan et al. (2021) | Deep learning, 1-D Residual Neural Networks | Apnea-ECG | ECG signals | 93.05% |
| Nassi et al. (2022) | Deep learning, WaveNet | MGH and SSHS dataset | PSG inputs | 84% (AHI index calculation) |
| Chyad et al. (2022) | Deep learning, MVO, ANN | Own dataset | PSG inputs | 98.67% |
| Hedman et al. (2021) | RNN | PhysioNet Apnea-ECG | ECG signal | 91.7% |
| Ivanko, Ivanushkina & Rykhalska (2020) | Machine learning | PhysioNet resource | ECG | 98.7% (apnea and normal) |
| Li et al. (2018) | SVM and ANN | Apnea-ECG dataset | Single lead ECG | 85% (OSA event detection |
| Gutierrez-Tobal et al. (2019) | Machine learning AdaBoost, linear discriminants | Own dataset | SpO2 | 78.7% (severe apnea classification) |
| Álvarez et al. (2020) | Machine learning | Own dataset | SpO2 and airflow | 81.3% (four class) |
| Song et al. (2016) | Machine learning, Hidden Markov model | The apnea-ECG | ECG | 86.2% |
| Rodrigues et al. (2020) | Machine learning | MARS dataset | PSG inputs | 83% (specify) |
| Huang et al. (2020) | SVM | Own dataset | PSG inputs | 82% (AUC) |
| Stretch et al. (2019) | Random forest | Own dataset | PSG inputs | 46% (sensitivity) |
| Mencar et al. (2019) | Machine learning | Own dataset | Gas exchange inputs | 44.7% (AHI index to represent Apnea severity) |
| Lazazzera et al. (2021) | Machine learning | Own and different dataset for testing | PPG and SpO2 | 75.1% (apnea and hypopnea) |

(Continued)

| Table 1 (continued) | | | | |
| --- | --- | --- | --- | --- |
| Reference | Method | Dataset | Channels | Accuracy |
| *Papini (2022)* | Machine learning | Own dataset | Heart and sleep data | 86% (AUC mild/moderate/severe OSA) |
| *Surrel et al. (2018)* | Machine learning SVM | PhysioNet Apnea-ECG | Single channel ECG | 88.2% (max) |
| *Varon et al. (2015)* | Machine learning SVM | Apnea-ECG | Single channel ECG | 84.7% |
| *Sharma & Sharma (2016)* | Machine learning KNN | Apnea-ECG | Single channel ECG | 83.8% |
| *Dutta et al. (2021)* | Machine learning | Own dataset | PSG inputs | 86% (AHI) |
| This work | Deep learning + Machine learning- Two-tier model | Own dataset occupied from 50 patients | PSG inputs, Snoring, Arousal, Sleep Stages, SpO2 | 1- Sleep Disorder Detection (Apnea, Hypopne or Normal) 95.76% 2- Apnea Events Detection (Mixed Apnea, Central Apnea or Obstructive Apnea) 99.4% |

*Yang et al. (2022)* proposed a structure using one-dimensional resudial networks and single-channel ECG signal data for the detection of sleep apnea. They achieved 90.3% success in their tests with the Apnea-ECG data. *Wang et al. (2022)*, proposed a model that can be used in IoT devices for sleep apnea monitoring, uses a single-channel EEG-based feature and classifies with Bi-LSTM.

Due to containing only one record where AHI is distributed near critical values of 5, some studies as in *Sharan et al. (2021)*, *Nassi et al. (2022)*, *Chyad et al. (2022)* achieves 100% accuracy in OSA detection. Their data contains small number of channels and algorithms have boosted dataset for increasing accuracy. In contrast to them our proposed method has an accuracy of 91% by using all channels rather than single or small number of inputs. Also, our proposed model is marking apnea event like, and sleep experts do. For this reason, following works explained but not given in Table 1. For example, *Sharan et al. (2021)* proposed a model in which single-channel ECG signal data are evaluated in a 1-D residual CNN network. *Nassi et al. (2022)* developed a DNN model on MGH dataset for binary classification of sleep apnea by applying boosting. Their accuracy obtained 100% accuracy for differentiation apnea and normal events. *Chyad et al. (2022)* proposed a complex model using neural network and soft computing algorithms for OSA estimation. Like our study, the model hybridly combines different sensor data such as heart rate, SpO2, chest movement, and thus has a high success rate of 98.67%.

## Machine learning based studies

On the other hand, the machine learning approach was preferred to provide the opportunity to evaluate more sensors together in detecting sleep disorders. It is mostly aimed to evaluate the PSG data. In this section, current studies using the machine learning approach are evaluated.

*Rodrigues et al. (2020)*, conducted a comparative study on the MARS dataset, including tests with 60 different classification models for AHI index calculation and OSA detection. They obtained 83% specificity by 60 algorithms (28 regressors and 32 classifiers for

attribute selection). *Huang et al. (2020)* developed an SVM model to predict AHI index in Chinese patients. Their model reached to maximum 82% AUC and 74.4% sensitivity. Home sleep apnea test model which was developed by *Stretch et al. (2019)* applied random forest model and reached a sensitivity of 46% in the classification of respiratory efforts ≤5 and ≥5. *Mencar et al. (2019)*, collected and analyzed data from 313 people with OSA. They used 19 different features together in their analysis. Their model yielded 44.7 accuracy level in prediction of AHI index to represent OSA severity. The accuracy level in this work was low due to choosing gas exchange as an input for model creation. Another important factor is unbalanced or limited data in dataset for training. *Lazazzera et al. (2021)*, proposed a model in which they used PPG and SpO2 sensor data together to detect sleep apnea and hypopnea, and they achieved 75.1% accuracy. *Papini (2022)* proposed a model that automatically estimate AHI with a deep learning model that uses the carddirespirotory and sleep information collected by a wrist worn IoT device as input. He obtained 86% max ROC-AUC value in mild/moderate/severe OSA (*Papini, 2022*). His model is based on deep learning methods applied on own dataset. *Surrel et al. (2018)* developed a time-domain analysis based embedded system to compute the sleep apnea score. They obtained 88.2% accuracy in the tests performed with the PhysioNet Apnea-ECG. *Varon et al. (2015)* detected sleep apnea with 84.7% accuracy using single-lead ECG data. *Sharma & Sharma (2016)* Using single-lead ECG data, they achieved 83.8% success with the machine learning model. *Song et al. (2016)* proposed a OSA detection approach based on ECG signal by using discriminative hidden Markov model and related algorithms with an accuracy of 86.2%. Corresponding parameter estimation algorithms are provided. *Dhruba et al. (2021)*, similar to our study, they developed a model that uses multiple sensor data such as ECG, heart rate, pulse rate, skin response, and SpO2 together in the diagnosis of OSA. *Dutta et al. (2021)*, conducted a study aiming to identify four different OSA types depending on the AHI index with unsupervised multivariate PCA analysis and data-intensive machine learning and achieved 86% success.

We prepared the following comparison table Table 1 which presents the obtained results, methods, and datasets of the proposed work with the similar works.

In summary, when the studies were examined as shown in Table 1, it was seen that Physionet's Apnea ECG dataset was used in the majority of the studies. In all studies that did not use image processing, features such as Ramp Peak Value, P-wave, T wave, QRS complex, R-R interval, P-R interval, S-T interval, Q-T interval derived from ECG continuous signal data were used. It has been seen that machine learning, CNN, LSTM models are used as classifiers and the classification performances vary between 46% and 98.7%. In all high-performance models, patients were classified as either Apnea or Not.

In this study both deep learning and machine learning approaches were used as a two-layer architecture design together. Own dataset was used in the training and testing processes of the design. While sleep disorder detection (apnea, hypopnea or normal) is performed in the first stage for classification, apnea's events (mixed, central, or obstructive) are also detected for patients which identified as apnea. In this aspect, it differs from other studies. Since they both determine disorders and events together. Classification performances are determined as 98.99% in the first stage and 99.4% in the second stage.

This high performance is depending on the two-layer architecture using and employing PSG inputs as input from supervised features (C-snore, desaturation, arousal, sleep stages). While the proposed model classifies with high performance, it will help to the sleep doctors or experts and support them for labor costs. This study is able to detect the disease as more classes of the person with the proposed model and does not use any feature engineering in it. Only sensor data are evaluated as features in model training.

## METHODOLOGY

In this study, a model in which a two-layer learning architecture is used, and high performance is achieved for the detection of sleep apnea types as multiple classes is proposed and is shown in detail in Fig. 1. Model includes data preparation, training and testing for meta-learner feature set creation with all data, meta-learner pre-training model and testing stages.

### Data preparation

One of the first and most important stages of the proposed 2-layer classification structure in this study is the preparation of the data set for classification and feature engineering. In this study, as in similar studies, used the PSG input (A1A2, ABD, Body, C3A2, C4A1, CEMG, CFlow, ECG2, F3A2, F4A1, LEG1, LEG2, LEOGA2, O1A2, O2A1, REOGA2, SpO2, TFlow, THO.), C_snore, de_saturation, arousal, and sleep stage features for modeling.

C_snore data is the feature that shows at which periods people snore during sleep and is kept as binary. Snoring, a type of respiratory sound, is one of the earliest and most common symptoms for the detection of sleep apnea syndrome (*Lin et al., 2022*). For this reason, it was evaluated as an important parameter in the classification of sleep disorders and was used in the model proposed in this study.

Oxygen saturation (SpO2) has been reported to facilitate the detection of OSA disease, especially in children (*Wu et al., 2022*). It has been shown that PSG data and SpO2 signal values are compatible with each other and OSA diagnosis can be made with artificial intelligence models in adults (*Li et al., 2021*).

*Gold et al. (2016)*, demonstrated the relationship between arousal status, sleepiness/fatigue, and AHI. It has been stated that sleepiness and fatigue are two symptoms that characterize the severity of the disease in OSA patients.

The symptoms of most sleep disorders can be determined objectively using the expert decision system. The physiological characteristics of these diseases are also directly related to the proportional change of sleep stages. For this reason, sleep staging is very important in terms of sleep health. According to the American Sleep Medical Academy, sleep stages are defined as Wake (W), REM, Non-Rem1, Non-Rem2, and Non-Rem3 (*Arslan et al., 2022*). Automatic sleep staging is important in OSA patients and provides important distinguishing data on patients. In addition, analysis of polysomnography (PSG) recordings containing data such as EEG, EOG, and EMG is also widely used in solving sleep-related problems (*Zhang et al., 2022*; *Arslan et al., 2022*; *Arslan et al., 2022*).

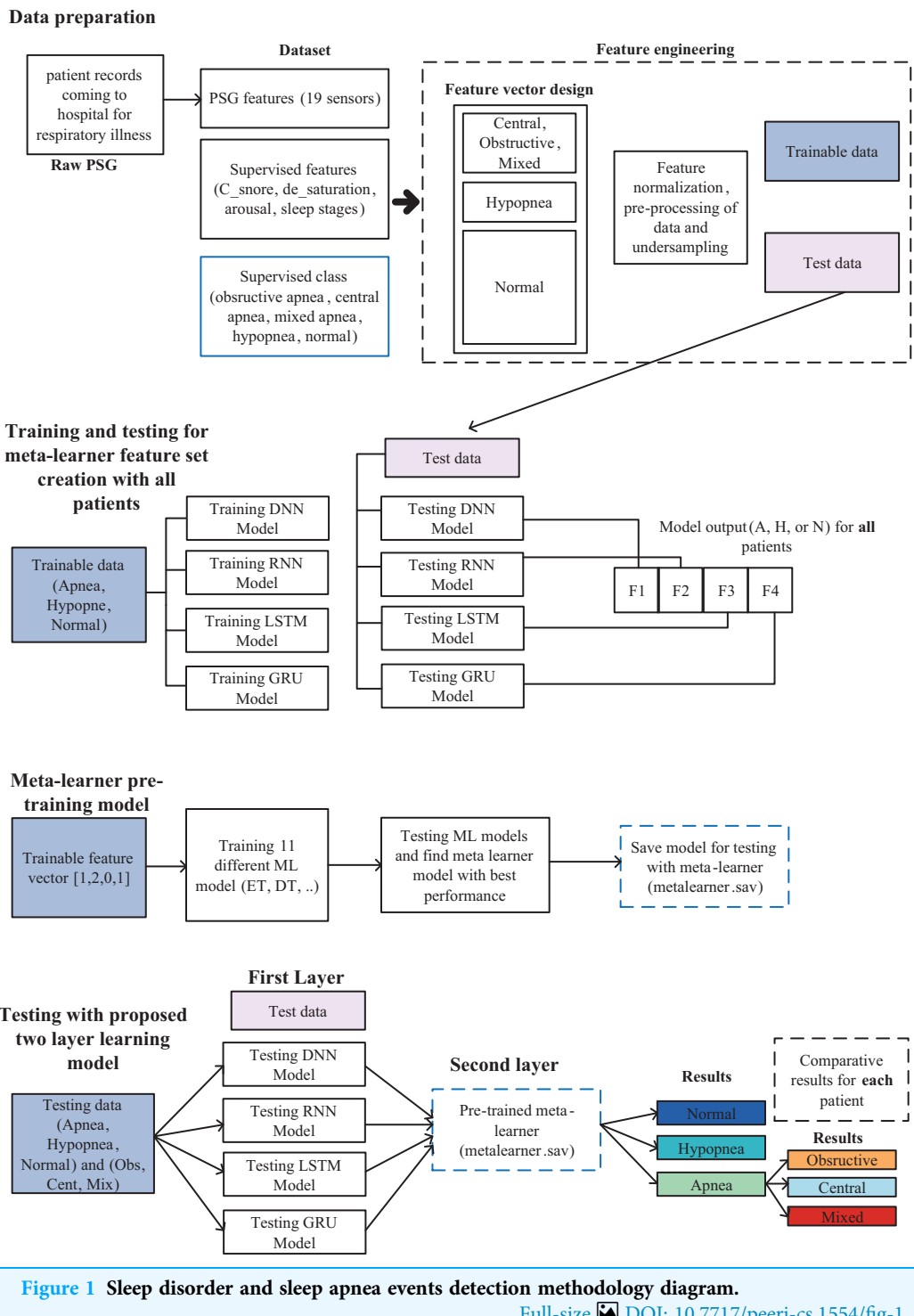

**Figure 1 Sleep disorder and sleep apnea events detection methodology diagram.**

As it is stated in the literature that different features contribute positively to classification, as given above, in the detection of sleep disorders and the determination of sleep apnea events, all of them were evaluated within the feature set in this study. A feature vector containing 23 features for each patient was prepared, along with 19 PSG records,

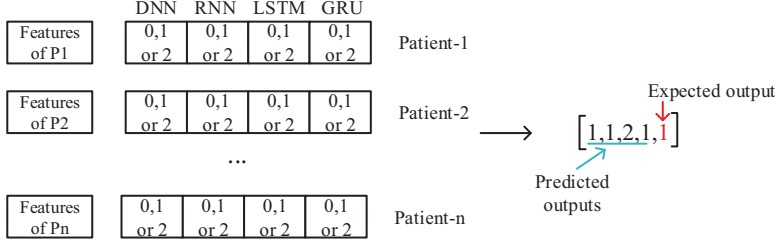

**Figure 2 Feature set design for training of meta-learner model.**

Csnore, SpO2, arousal status, and sleep score. Data were collected over a sleep period of approximately 8 h for each of the 50 patients. Due to the different nature and high number of features, a series of pre-processing steps were applied as shown in Fig. 1. These operations are normalization of data, elimination of sample imbalance between clusters, cleaning of erroneous and NaN data. Necessary ones of these procedures were applied for each patient. As a result, a usable feature set was obtained in the two-layer architecture proposed for this study. The data obtained as a result of these processes were splitted into 70% training and 30% testing. A total of 10% of the training set was used as validation.

## Meta-learner feature set creation and pre-trained model design

The meta-learner design is made, after the data preparation is completed. The collected data for 50 patients are separated as training and test sets. With the training data, DNN, RNN, LSTM and GRU deep learning models are trained and tested with the test data and the results are recorded. The output of each model is to determine one of the three classes as Apnea, Hypopnea or Normal. As shown in Fig. 2, a vector containing five features with four deep learning models using class prediction as input and one with expected output is created. As a result, a feature vector of 3,077,200 × 23 dimensions is obtained by combining the data of 50 patients into a single feature vector.

This generated intermediate dataset is used in the training of the meta-learner model. An important point at this point is to evaluate the data of all 50 patients together in a single feature vector. Thus, a patient-independent second layer architecture is created. This model will then be used to evaluate the results, working for each patient individually. Since the patient's age, gender, and other diseases affect sleep disorders, a patient-based evaluation is required. However, with the proposed model, a patient-independent two-tier classifier is achieved. The same trained model is used for testing of all patients.

## Proposed two-layer classification model

After all the pre-processing stages are completed, a model is revealed, in which both sleep disorders and sleep apnea events can be detected. Although it seems like a disadvantageous situation that the preprocessing and model preparation phase is created in a few steps and it is a relatively complex model, it is possible to make a stable and high-performance classification that does not change from patient to patient. While the model determines sleep disorders as multiclass as apnea, hypopnea and normal, it can detect apnea events as obstructive, mixed, and central.

**Table 2 Dataset details.**

| PatientID | Before undersampling data size | | | | After undersampling data size | | | |
|---|---|---|---|---|---|---|---|---|
| | Apnea | Hypopnea | Normal | Total | Apnea | Hypopnea | Normal | Total |
| Total data | 761,000 | 1,843,200 | 226,651,000 | 229,255,200 | 761,000 | 1,843,200 | 617,000 | 3,221,200 |
| Average | 15,220 | 36,864 | 4,533,020 | 4,585,104 | 15,220 | 36,864 | 12,340 | 64,424 |

## Dataset

The dataset was provided by Yozgat Bozok University, Department of Chest Diseases Sleep Laboratory by getting necessary ethical permissions. As will be given Annex 1, 50 patients' data were used in the training and testing processes of this study. While there were 761,000 apnea records for 50 patients, there are 1.8 million hypopnea records, and 226 million normal records were deducted in Table 2. Even if a person has very severe apnea, he shows signs of apnea or hypopnea for an 8-h sleep period are relatively smaller than the normal situation. This situation negatively affects the training of the model in both learning and classification stages and causes to the tendency of the learned model to mark all situations as normal in the test phase. To resolve severe sample imbalance problem between clusters, undersampling was performed according to the "majority" class and normally marked samples were reduced for each patient separately. As a result, the number of normal cases, which was 4.5 M per patient on average, was reduced to 12 thousand. Thus, sample imbalance between clusters was eliminated.

This study is realized by using large amount of data which can be seen in the number of samples before under-sampling and results are compared with 50 patients one by one. Thus, it is possible to evaluate the conditions such as age, gender, other existing diseases, and measurement errors. Each patient was evaluated individually in his or her own situation.

## Deep learning model structures and machine learning models

The deep learning structures in four different architectures which will be used in meta-learner training of the proposed model and testing results proposed in this study and shown below.

Traditional neural networks have only one hidden layer. For this reason, they are easily trained but have difficulty solving complex problems. One of the networks used in this study is the deep neural network (DNN) since the detection of sleep disorders is also very complex and varies according to the patient's condition. The most important advantage of this network is that it has a deep architecture and allows deep features to be learned by having more hidden layers. Considering that there are approximately 4.5 M records with 23 features per patient, it can be thought a set of features may be extracted.

RNN (*Sherstinsky, 2020*) is not only fully connected between adjacent network layers, but also interconnected at neurons in each layer. This structure allows information to be

transmitted between neurons, and the output of each neuron acts as the input of the next neuron. While he is good at learning about short-term addictions, he has a hard time remembering long-term addictions. Since sleep data was produced during sleep, RNN model could be successful in detecting sleep apnea by examining short-term dependencies.

Unlike RNN, LSTM (*Hochreiter & Schmidhuber, 1997*) is better at learning long-term dependencies. This model consists of inputs, LSTM, full link layer and output layer. It uses a kind of transitive structure to capture long-term dependencies between data to prevent data losses. Since it was considered that it would be possible to detect sleep disorders by taking into account the long-term dependence of sleep data over time, it was used among the deep learning architectures used in this study.

GRU (*Cho et al., 2014*) can be thought of as a simplified version of the LSTM structure by making some transitions. It has fewer parameters than LSTM because it has no output gate. The GRU update port determines how much of the previous data will be remembered, while the reset port decides how to combine the previous data with the current data. In this study, the GRU structure was also evaluated between test environments to see the comparative results and to evaluate the dependence of sleep disorders on sleep recordings during the sleep process. The purpose of evaluating so many different structures together is to analyze data dependencies in the best way and to achieve the highest performance.

To observe the state of addiction in sleep data and to reveal the most successful mode among the learning structures, four different learning structures were designed as summarized in Fig. 3. Their test results are shown and analyzed by comparing them in next section.

In deep learning structures, there are basically input layer, hidden layers, and output layers. Structure parameters can be listed as size (total number of nodes), width (number of nodes in the relevant layer), depth (number of layers in the network), capacity (learning function structure and type) and architecture (layer layout of the network). These parameter values are directly affecting the system performance and there is no global system proposal in determining these values. So that, an empirical approach has been adopted because of the experiments to ensure that the most successful model was revealed.

After the training and testing processes of the four deep learning structures given above, some problems were identified. To solve these problems, a meta-learner is used in the second layer proposed in this study. Machine learning models were used in this decision-making layer. The reason is to obtain more stable values in the results.

For the meta-learner design, tests were carried out with 11 different machine learning algorithms, namely logistic regression (LR), random forest (RF), decision tree (DT), Gaussian naive bayes (GNB), linear discriminant analysis (LDA), Ada Boost, gradient boosting (GB), ExtraTree (ET), Extreme Gradient Boosting (XGBoost), support vector classifier (SVC), K-nearest neighbor (KNN). Thus, it is aimed to select the machine learning model with the highest performance and use it as a decision model. This allows us to guarantee the highest classification success for all patients. Thus, the proposed model not only aims to increase performance, but also eliminates patient-based variability in

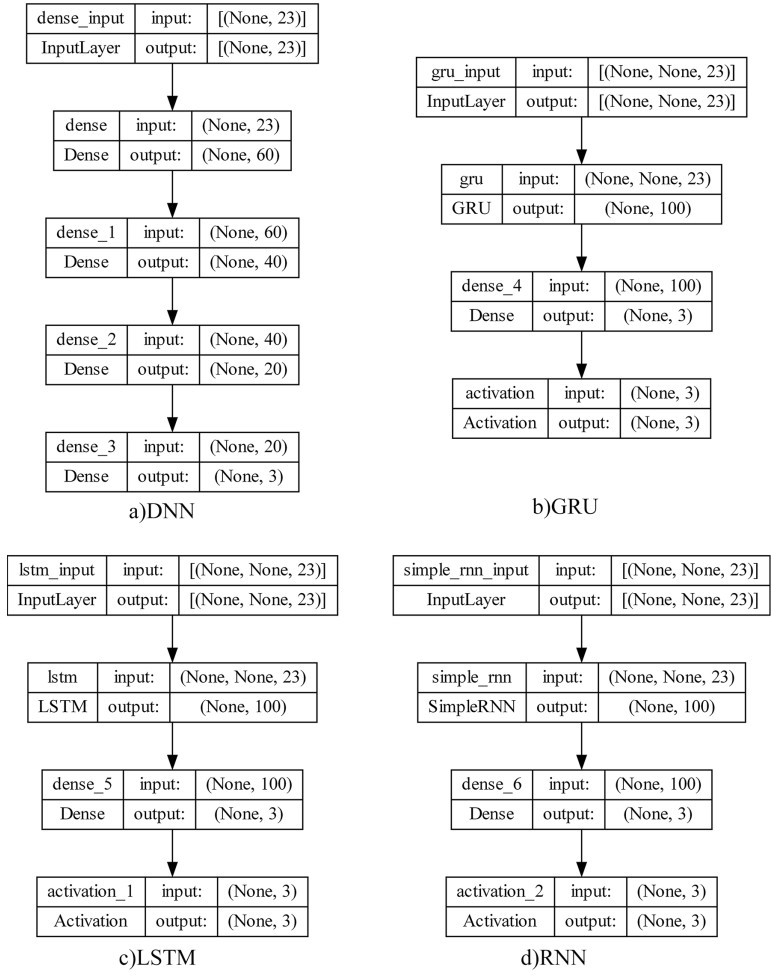

**Figure 3  Deep learning model architectures.**

results. Default hyper-parameters were used for each classifier and no parameter optimization was performed.

# OBTAINED RESULTS

## System architecture

Training of both deep learning models, meta-learner's training and all test processes performed with a notebook equipped with 32 GB RAM and Intel(R) Processor 11th Gen Intel(R) Core (TM) i7-11390H @ 3.40 GHz, 2918 Mhz, four Cores, eight Logical Processors. The server has Windows 10 operating system. Tensorflow is used with Keras framework. In addition, matplotlib, sklearn, imblearn, numpy, pandas' libraries were used. Different libraries were needed for the two-layer model with both deep learning and machine learning structures.

## Evulation

Traditional performance measurement parameters of accuracy, specificity, recall, f-score and confusion matrix were used for calculation, evulation and comparison of results both

in deep learning models and machine learning models. The calculation equations of these parameters are as shown below.

$$\text{Accuracy} = \frac{TP + TN}{TP + TN + FP + FN} \qquad \text{Sensitivity} = \frac{TP}{TP + FN}$$

$$\text{Specificity} = \frac{TN}{FP + TN} \qquad \text{F-score} = \frac{2 \times TP}{2 \times TP\ FN + FP}$$

$$\text{Confusion matrix} = \begin{bmatrix} TN & FP \\ FN & TP \end{bmatrix}$$

## Deep learning models for sleep disorder detection

The learning curves for each deep learning model were as given below in order to obtain the best results for each model on a patient basis and to understand that the learning process is now being completed.

The learning curve for four different models was as shown in Fig. 4. When the graphs are examined, it is seen that the training and test curves are parallel and could achieve high performance. However, the amount of vibration could not be reduced, and smoother graphics could not be obtained. When we set the batch size to be larger, smoother curves emerge, but in this case, there is a risk of fluctuation. Since the learning rate also affects this situation, the learning rate was tried to be determined in a balanced way with the batch size. On the other hand, the smoothness of the curve is not seen as a problem. Because batch size and learning rate changes do not cause much fluctuations on graphics. This showed that the proposed model did not have any major problems with convergence or overfitting. The number of epochs, on the other hand, varies according to the complexity and structure of the model, and the training was terminated when there was no longer any increase in classification performance.

Since the problem examined in this study is multi-class (Apnea, Hypopnea or Normal), the results obtained for 6 different patients are shown in Fig. 5 as below to evaluate the high performance obtained in the learning curves on a class basis. It has been observed that TP values are generally high and average performances are 95% and above. However, a problem arose in the results obtained as can be seen in the figure. This problem is detection of higher performance in Apnea or Hypopnea class changing patient to patient. This situation revealed that deep learning models can show different performances, varying from patient to patient. It causes patient-to-patient variability in the detection of positive patients expressed as Recall. It tells that the results are balanced on a class basis according to the confusion matrices and that similar results cannot be guaranteed for each sleep disorder. This limits the stability of the proposed model and its ability to produce patient-independent results. For this reason, it is aimed to solve this problem by using a meta-learner in layer 2.

Another situation that should be evaluated in tests with deep learning models is defining which model produces more successful results on a patient basis. As this study aimed to design a patient-independent system, it is expected that the results will be achieved with the highest performance for each patient. The highest performance model table which was obtained for the 50 patients used in this study is given in Table 3. Accordingly, while the

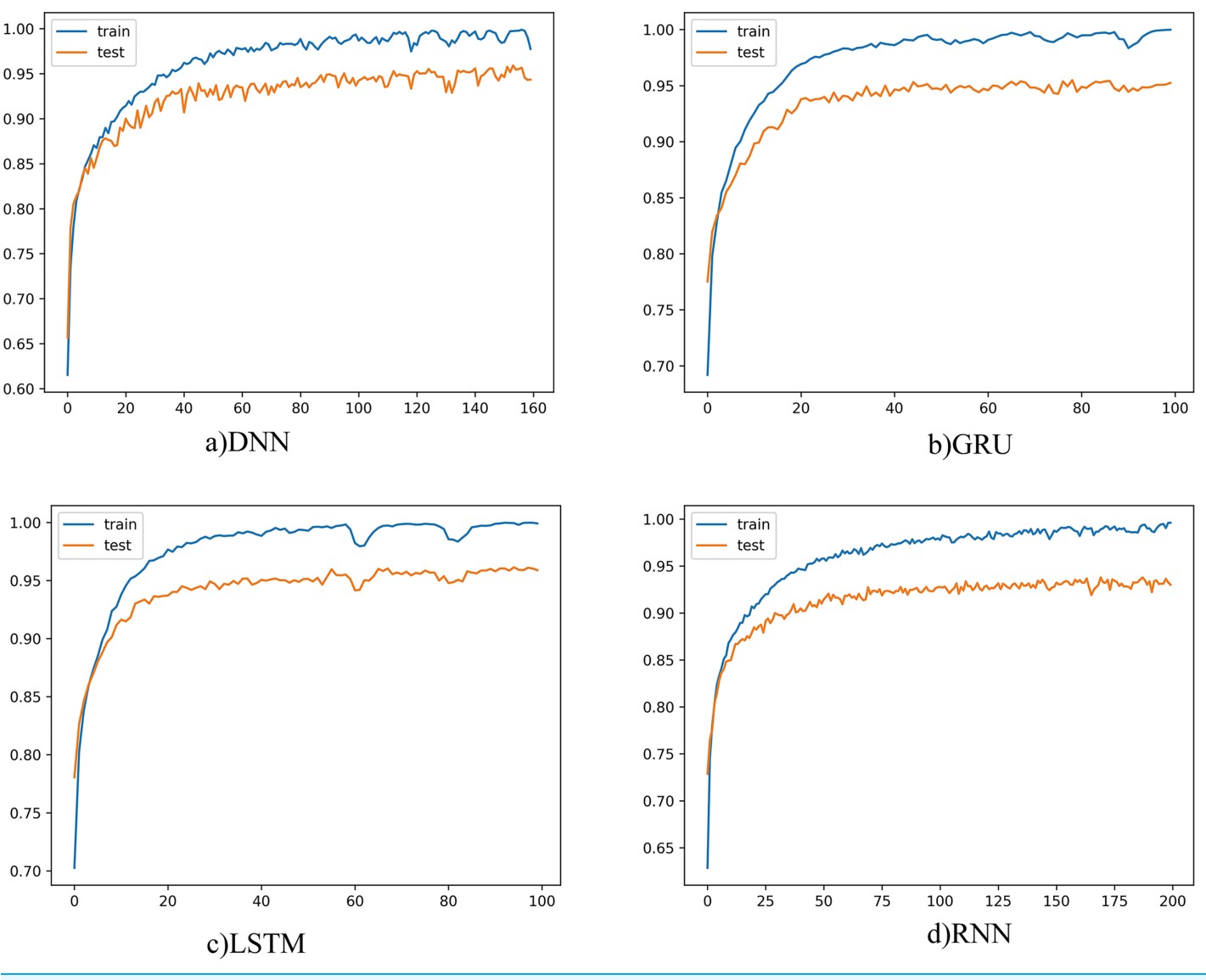

**Figure 4 Deep learning models learning curves.**

DNN model has the highest performance in most patients, there are patients with higher performance in the LSTM, GRU and RNN models. Although there are not high differences in classification performances as can be seen in all results given in Annex-2, it is planned to be examined and resolved in this case. To do so, evaluation of each model's output with a meta learner has been completed to ensure the best result. In this way, this work not only achieve an increase in performance, but more importantly, ensures that the results remain consistent from patient to patient. As can be seen in the table, the highest results in all patients were obtained with the proposed two-layer architecture as given in Annex-2.

As a result, when the results obtained in this section and some limitations of deep learning models are evaluated together, the following problems emerged:

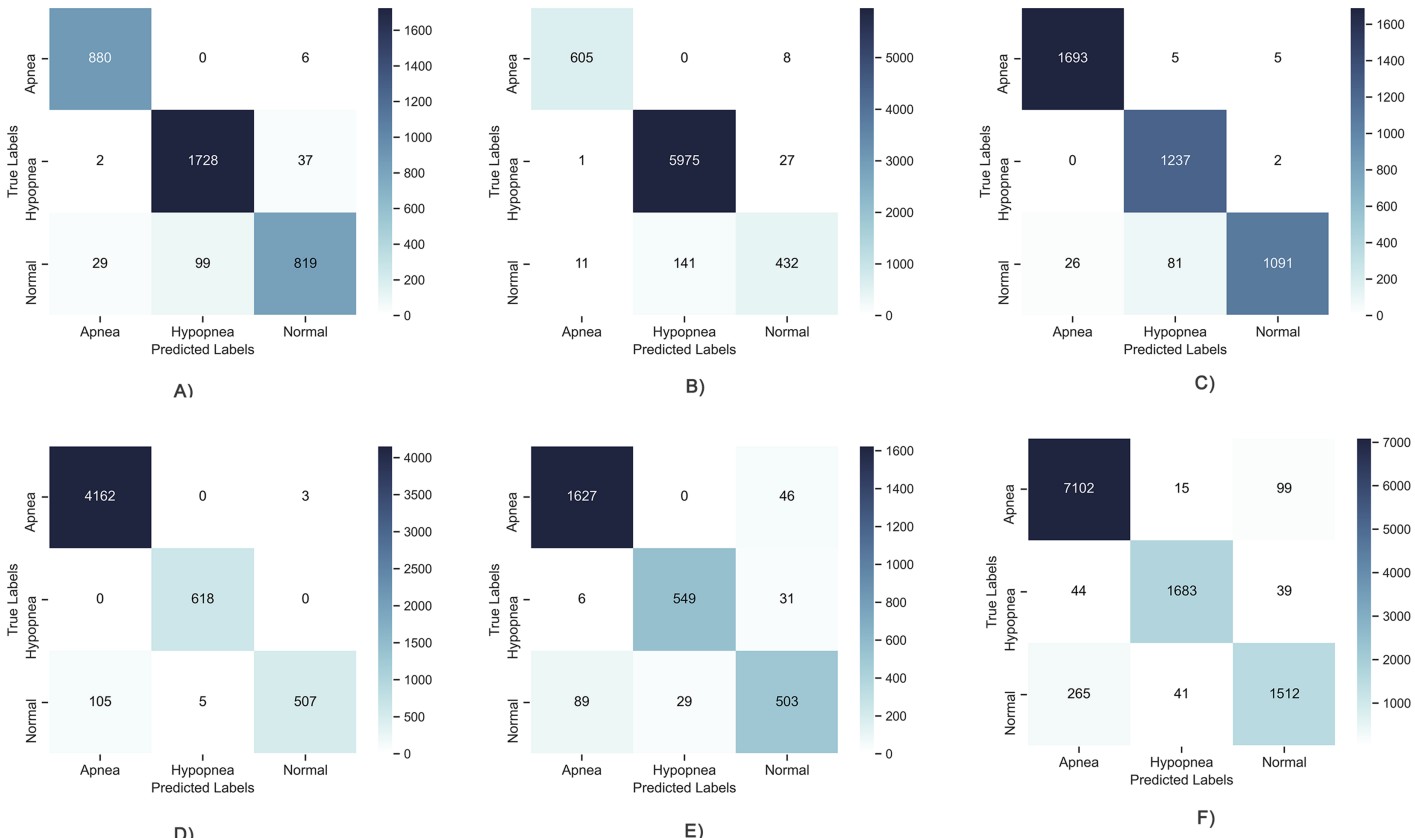

**Figure 5 Confusion matrix of deep learning models.** (A) Patient 4, Apnea 98.97%, Hypopnea, 96.17%, Normal, 92.25%; (B) Patient 12, Apnea 99.72%, Hypopnea, 97.65%, Normal, 97.40%; (C) Patient 5, Apnea 99.15%, Hypopnea, 97.92%, Normal, 97.30%; (D) Patient 7, Apnea 98.92%, Hypopnea, 99.95%, Normal, 98.87%; (E) Patient 1, Apnea 95.1%, Hypopnea, 97.71%, Normal, 93.23%; (F) Patient 22, Apnea 96.08%, Hypopnea, 98.71%, Normal, 95.89%.

1- As can be seen with confusion matrixes, different sleep disorders are more successful for different patients. This disrupts the consistency of deep learning models. The level of success may rise or fall in certain types of disease on a patient basis.

2- There is a problem that different deep learning models are more successful in different patients and therefore it is not possible to decide which model to use and the best result cannot always be guaranteed.

3- It is necessary to evaluate whether it is possible to increase the classification performance.

4- Since the age, gender and status of other diseases are effective in sleep apnea, results that vary from patient to patient can be obtained.

It has been evaluated that these problems can be solved by using a meta-learner in the 2nd layer to solve these problems. For these reasons, a two-layer architecture was designed, and it was aimed to find solutions to these problems and increase the overall system performance. The results obtained with the use of meta learner are explained in the next section.

**Table 3 Best learning models for each patient.**

| | Patients with the maximum classification result achieved | Proposed two-tier model |
|---|---|---|
| DNN | 2, 4, 7, 9, 10, 12–19, 21, 23, 24–31, 33–40, 42, 43, 45, 47, 49, 50 | |
| LSTM | 1, 5, 6, 8, 20, 32, 44, 46, 48 | It has higher performance than the model with the best performance in all patients. |
| GRU | 11, 22, 41 | |
| RNN | 3 | |

## Results for meta-learner preparation

One of the important features of this study is to analyze the results in a second layer and increasing the model performance. At this stage, it is aimed to eliminate the limitations detailed in the previous section. For this purpose, the data of all patients were collected in a single dataset and tested with 11 different machine learning models. By choosing the classifier with the best results, its contribution to the results seen positively. After these tests, the results have reached very high accuracy changing between 82.0% to 99.5%. Furthermore, these results showed that the ensemble models such as ExtraTree, XGBoost, Gradient Boosting are more successful than other types. For this reason, one of them is trained as a meta-learner to be used in the final stage of the model. Thus, the decision-making layer that will take four different deep learning model inputs as features to produce a three-class output has been prepared.

Training is performed for four classification results and one expected value after under-sampling for all patients, and the result for all classifiers is as shown in Fig. 6.

As a result, the ExtraTree classifier with the highest success is chosen for the meta-learner. Whether this selection contributes to the classification is evaluated by testing in the next section.

## Results for two-tier proposed model (meta learner design) for sleep disorder detection (apnea, hypopnea, normal)

To compare the results of the proposed model with the standard deep learning models, the same data has been tested with both deep learning models and with the proposed two-layer architecture, obtained results are presented in the table shown below seperately in Table 4.

When the average, maximum and minimum results of DNN, LSTM, GRU, RNN and the proposed model (PM) in this study obtained for 50 patients is examined, it is seen that the PM haas higher success than other classifiers in terms of all parameters. Average performance increase is 1.19%. In addition, the lowest performance value has increased to 90.33%, while the highest performance has reached very high values such as 98.99%. The average performance has also been increased to over 95%. This increase is similar for all metrics, proving that the proposed model makes this contribution regardless of the class. To better observe this contribution, confusion matrixes obtained for the patient with the lowest performance are given in Fig. 7.

When the matrixes are examined, it is seen that the sample numbers for the apnea, hypopnea and normal classes are balanced at the first stage. On a class basis, the highest

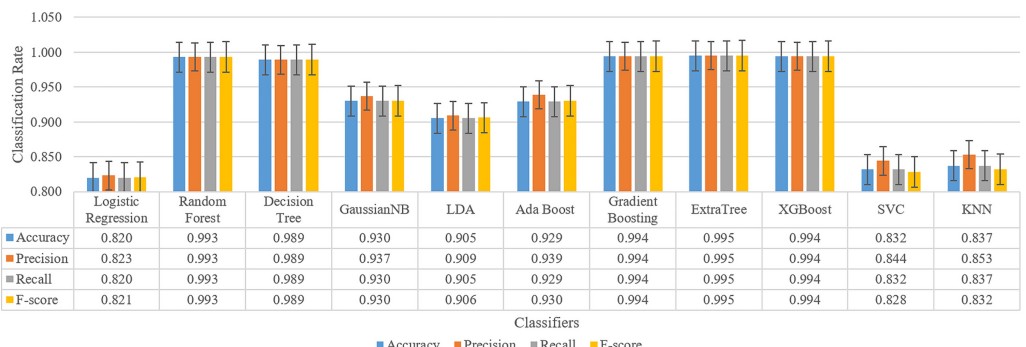

**Figure 6 Classification results for sleep disorder detection (apnea, hypopenea, normal) for meta-learner design.**

**Table 4 Classification results with proposed model.**

|  |  | Accuracy | Precision | Recall | F-score |
|---|---|---|---|---|---|
| DNN | Average | 0.9457 | 0.9456 | 0.9457 | 0.9447 |
|  | Min | 0.8878 | 0.8868 | 0.8878 | 0.8848 |
|  | Max | 0.9847 | 0.9848 | 0.9847 | 0.9845 |
| LSTM | Average | 0.9384 | 0.9376 | 0.9384 | 0.9372 |
|  | Min | 0.8457 | 0.8432 | 0.8457 | 0.8440 |
|  | Max | 0.9862 | 0.9864 | 0.9862 | 0.9862 |
| GRU | Average | 0.9337 | 0.9329 | 0.9337 | 0.9325 |
|  | Min | 0.8392 | 0.8377 | 0.8392 | 0.8380 |
|  | Max | 0.9815 | 0.9818 | 0.9815 | 0.9814 |
| RNN | Average | 0.9170 | 0.9162 | 0.9170 | 0.9151 |
|  | Min | 0.7884 | 0.7871 | 0.7884 | 0.7860 |
|  | Max | 0.9838 | 0.9844 | 0.9838 | 0.9836 |
| Proposed two-layer learning model. | Average | 0.9576 | 0.9574 | 0.9576 | 0.9568 |
|  | Min | 0.9033 | 0.9019 | 0.9033 | 0.9022 |
|  | Max | 0.9899 | 0.9899 | 0.9899 | 0.9898 |
| Avg. performance increase |  | 1.19% | 1.17% | 1.19% | 1.12% |

value in all metrics in all three classes was obtained with the proposed model. The closest results to the proposed model are obtained with the DNN model. The accuracy value increased by 2% compared to the closest classifier.

In this section, the contributions of the proposed model to the performance increase compared to other deep learning models are presented together with the test results. Other contributions of the proposed model (1) were created a structure that did not change from patient to patient and all patients' data were evaluated on a single model, (2) all patient data were used in a single model training. Thus, all the conditions affecting the sleeping sickness such as the patient's (age, gender, other diseases) were evaluated for a single model. As a result, a high-performance sleep disorder detection tool was developed with a model that eliminates some limitations independently of the patient, has four different

| DNN | Precision | Recall | F1-score | Support |
|---|---|---|---|---|
| Apnea | 0.94 | 0.93 | 0.93 | 4065 |
| Hypopnea | 0.93 | 0.94 | 0.94 | 6015 |
| Normal | 0.88 | 0.88 | 0.88 | 4080 |
| | | | | |
| Accuracy | | | 0.92 | 14160 |
| Macro avg | 0.92 | 0.92 | 0.92 | 14160 |
| Weighted avg | 0.92 | 0.92 | 0.92 | 14160 |

| GRU | Precision | Recall | F1-score | Support |
|---|---|---|---|---|
| Apnea | 0.91 | 0.93 | 0.92 | 4065 |
| Hypopnea | 0.93 | 0.94 | 0.93 | 6015 |
| Normal | 0.88 | 0.84 | 0.86 | 4080 |
| | | | | |
| Accuracy | | | 0.91 | 14160 |
| Macro avg | 0.91 | 0.90 | 0.90 | 14160 |
| Weighted | 0.91 | 0.91 | 0.91 | 14160 |

| LSTM | Precision | Recall | F1-score | Support |
|---|---|---|---|---|
| Apnea | 0.89 | 0.93 | 0.91 | 4065 |
| Hypopnea | 0.92 | 0.93 | 0.93 | 6015 |
| Normal | 0.88 | 0.83 | 0.85 | 4080 |
| | | | | |
| Accuracy | | | 0.90 | 14160 |
| Macro avg | 0.90 | 0.90 | 0.90 | 14160 |
| Weighted | 0.90 | 0.90 | 0.90 | 14160 |

| RNN | Precision | Recall | F1-score | Support |
|---|---|---|---|---|
| Apnea | 0.84 | 0.92 | 0.88 | 4065 |
| Hypopnea | 0.89 | 0.91 | 0.90 | 6015 |
| Normal | 0.85 | 0.74 | 0.79 | 4080 |
| | | | | |
| Accuracy | | | 0.86 | 14160 |
| Macro avg | 0.86 | 0.86 | 0.86 | 14160 |
| Weighted | 0.86 | 0.86 | 0.86 | 14160 |

| Proposed Model | Precision | Recall | F1-score | Support |
|---|---|---|---|---|
| Apnea | 0.95 | 0.95 | 0.95 | 4065 |
| Hypopnea | 0.95 | 0.95 | 0.95 | 6015 |
| Normal | 0.91 | 0.90 | 0.90 | 4080 |
| | | | | |
| Accuracy | | | 0.94 | 14160 |
| Macro avg | 0.93 | 0.93 | 0.93 | 14160 |
| Weighted avg | 0.94 | 0.94 | | 14160 |

**Figure 7  Detailed confusion matrix with three class basis.**

deep learning models in the first layer, and Extra Tree ensemble classifier in the second layer.

## Results for apnea events detection (obstructive, mixed or central apnea)

Sleep apnea refers to situations in which a person's breathing stops and pauses intermittently throughout the night. Obstructive apnea is the most common form of sleep apnea. However, it has two different forms as central or mixed sleep apnea. In the second test phase of the proposed model, the events of patients with Apnea type are also determined. It has been tested with both machine learning models and deep learning models to detect three types of sleep apnea, and the results are as shown in Table 5. Like the previous results, ensemble models showed the highest performance in this problem. Events of apnea patients are classified with a high accuracy of 99.4%. Precision, recall and f-score values are also at a similar level. The proposed model showed high performance in this problem.

Comparison results are given in Fig. 8 to evaluate the classification results on ROC curve and confusion matrix. At this point, it is necessary to evaluate the results on a class basis for a multi-class problem.

In Fig. 8A, sleep apnea events were tested in three classes as central, mixed and obtrusive. It seen on the tests that there are only nine FP or FN samples over 1,800 samples. Among these, central apnea and mixed apnea cases can be detected with much higher

**Table 5 Results for apnea events detection.**

| Algorithm | Type | Classification algorithm | Accuracy | Precision | Recall | F-score |
|---|---|---|---|---|---|---|
| Machine learning | Regression algorithm | Logistic Regression | 0.801 | 0.807 | 0.801 | 0.802 |
| | Decision Tree | C 4.5 | 0.984 | 0.984 | 0.984 | 0.984 |
| | Bayesian | GaussianNB | 0.924 | 0.932 | 0.924 | 0.924 |
| | LDA | LDA | 0.911 | 0.916 | 0.911 | 0.911 |
| | Ensemble | Ada Boost | 0.931 | 0.937 | 0.931 | 0.931 |
| | | Gradient Boosting | 0.990 | 0.990 | 0.990 | 0.990 |
| | | Random Forest | 0.987 | 0.987 | 0.987 | 0.987 |
| | | ExtraTree | 0.994 | 0.994 | 0.994 | 0.994 |
| | | XGBoost | 0.993 | 0.993 | 0.993 | 0.993 |
| | Instance-based | SVC | 0.839 | 0.848 | 0.839 | 0.836 |
| | | KNN | 0.841 | 0.858 | 0.841 | 0.837 |
| Deep learning | DNN | | 0.970 | 0.970 | 0.970 | 0.970 |
| | GRU | | 0.980 | 0.980 | 0.980 | 0.980 |
| | LSTM | | 0.970 | 0.970 | 0.970 | 0.970 |
| | RNN | | 0.981 | 0.981 | 0.981 | 0.981 |

performance, while there are erroneous detections in obstructive apnea cases. This high performance is also seen in the ROC curve. While the AUC value was 1.00 for both Central and Mixed Apnea, it was 0.99 for Obstructive Apnea. This shows that the proposed model can detect the events of Apnea patients with high performance.

The results of the tests performed to evaluate both sleep disorders and the events of Apnea patients together are given in Fig. 8B. At this stage, it is aimed to analyze the entire study together. Accordingly, high TP values were obtained except for the patients in the Normal state. "Normal" patients are confused with hypopnea patients. This situation is considered as problematic, and the proposed model is working in two stages. At the first stage, Apnea status of the patient will be determined and then its events will be determined.

## Hyper-parameters of proposed deep learning models

The four-layer DNN model consists of an input layer, two hidden layers, and an output layer. The input layer represents the data entering the model, and each input data in this model consists of 23 features. This layer has 60 neurons and ReLU (rectified linear unit) activation function. ReLU is often used in hidden layers because it converts negative input values to zero while leaving positive input values intact. Hidden layers enable the model to model complex relationships and patterns between the input and output layers. The hidden layers have 40 and 20 neurons in this model, respectively, and both use the ReLU activation function. More neurons and layers often add to the complexity of the model and often to its learning capacity. The output layer represents the output or prediction of the model. In this case, the model solves a multi-class classification problem. Therefore, three neurons are used in the output layer and Softmax is used as the activation function. To measure the success of the model, categorical_crossentropy is used as the loss function.

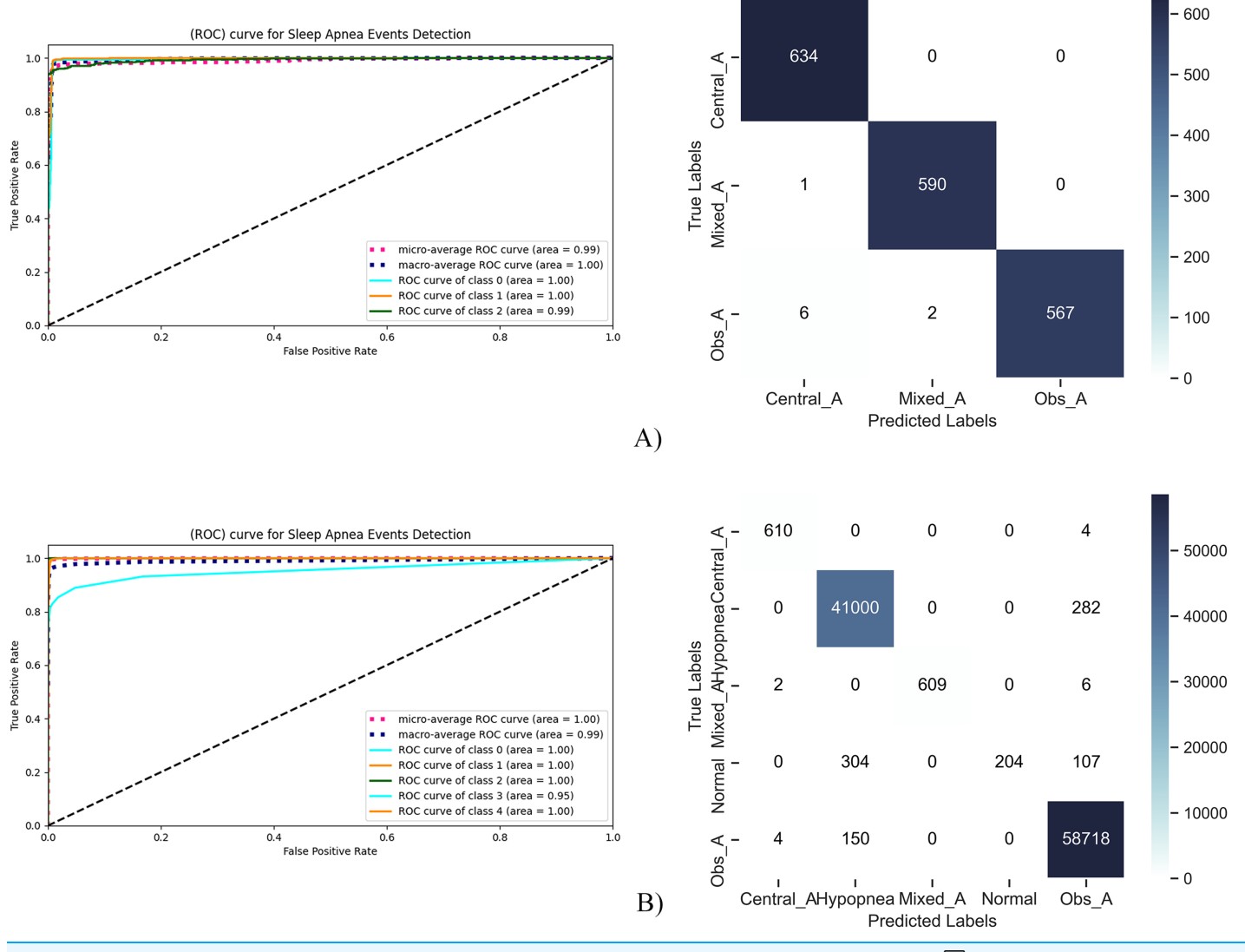

**Figure 8 RoC curve and confusion matrix for sleep apnea events detection.**

This loss function measures the difference between the actual values and the model's predictions. In addition, the Adam optimization algorithm is used to update the weights of the model. Adam adapts the learning rate and generally offers fast learning and good performance. The parameters selected for each deep learning model are as given in Table 6 with the tuning intervals.

## DISCUSSION

### Comparison with previous works

In this study, a two-layer structure was created that includes both deep learning and machine learning models. Besides PSG data, SpO2, Snoring, Arousal and Sleep Scoring data were used as features. When similar studies are examined (Table 7), some studies use a single signal data such as ECG signal, while others use more features consisting of PSG

**Table 6 Hyper-parameter tuning results.**

| Parameter | Tuning range | Selected parameters for each DL models |
|---|---|---|
| Kernel_initializer | Uniform, lecun_uniform, normal, zero, glorot_normal, glorot_uniform, he_normal, he_uniform | DNN: uniform |
| | | GRU: uniform |
| | | LSTM: uniform |
| | | RNN: uniform |
| Optimizer | SGD, RMSProp, Adagrad, Adadelta, Adam, Adamax, Nadam | DNN: adam |
| | | GRU: Nadam |
| | | LSTM: adam |
| | | RNN: adam |
| Learning_rate | (0.0001, 0.0005, 0.0008, 0.001, 0.01, 0.2, 0.3) | DNN: 0.0001 |
| | | GRU: 0.0002 |
| | | LSTM: 0.0001 |
| | | RNN: 0.0008 |
| Batch_size | (5, 10, 20, 30, 40, 50, 60, 70, 80, 90, 100) | DNN: 20 |
| | | GRU: 5 |
| | | LSTM: 5 |
| | | RNN: 10 |
| Epoch | (10, 20, 50, 100, 150, 160, 170, 200, 300) | DNN: 160 |
| | | GRU: 100 |
| | | LSTM: 100 |
| | | RNN: 200 |
| Neuron activation function | Softmax, softplus, softsign, relu, tanh, sigmoid, hard_sigmoid, linear | DNN: softmax |
| | | GRU: softmax |
| | | LSTM: sigmoid |
| | | RNN: softmax |
| Number of neurons | (1, 5, 10, 15, 20, 50, 100, 200, 500) | DNN: 60, 40, 20 |
| | | GRU: 100 |
| | | LSTM: 100 |
| | | RNN: 100 |
| Early stopping patient | | 20% |

inputs. Classification performances vary depending on the dataset and the number of classes. Deep learning models generally have higher performance than machine learning models. As a dataset, PhysioNet Apnea-ECG was used in most of the studies, while original datasets were created in some studies, as in our study.

In terms of model classification performance of proposed model; it detects sleep disorders with 95.76% accuracy and detects sleep apnea events with 99.4% accuracy. The first difference of our study from similar studies is that high performance was achieved for a multiclass problem. In previous studies, classification problems such as Apnea-Hypopnea, Apnea-Normal are generally solved in binary. By using 23 different features, more distinctive features of the patient were captured. In addition, our model can solve two different problems at the same time. It first detects the patient's sleep disorder, and in the

**Table 7 Similar works.**

| Reference | Method | Dataset | Channels | Accuracy |
|---|---|---|---|---|
| Erdenebayar et al. (2019) | 1D CNN and 2D CNN, RNN | Samsung Medical Center | ECG | 99% |
| Wang et al. (2019) | Deep learning, LeNet-5 | PhysioNet Apnea-ECG | Single Channel ECG | 87.6% |
| Hedman et al. (2021) | LSTM | PhysioNet Apnea-ECG | ECG | 97.1% |
| Chyad et al. (2022) | Deep learning + MVO | Own dataset | PSG inputs | 98.67% |
| Hedman et al. (2021) | RNN | PhysioNet Apnea-ECG | ECG signal | 91.7% |
| Rodrigues et al. (2020) | Machine learning | MARS dataset | PSG inputs | 83% (specify) |
| Huang et al. (2020) | SVM | Own dataset | PSG inputs | 82% (AUC |
| Stretch et al. (2019) | Random forest | Own dataset | PSG inputs | 46% (sensitivity) |
| Lazazzera et al. (2021) | Machine learning | Own and different dataset for testing | PPG and SpO2 | 75.1% (apnea and hypopnea) |
| Surrel et al. (2018) | Machine learning, SVM | PhysioNet Apnea-ECG | Single channel ECG | 88.2% (max) |
| Dutta et al. (2021) | Machine learning | Own Dataset | PSG inputs | 86% AHI = 56% accuracy |
| This work | Deep learning | Own dataset occupied from 50 patients | PSG inputs, Snoring, Arousal, Sleep Stages, SpO2 | 1- Sleep Disorder Detection (Apnea, Hypopne or Normal) 95.76% 2- Apnea Events Detection (Mixed Apnea, Central Apnea or Obstructive Apnea) 99.4% |

second stage, it can detect the events of apnea patients. In fact, it can detect for six different classes. In this aspect, it is completely different from other studies.

## Data imbalance problem

In multi-class problems, excessive imbalance between classes leads to an imbalance in the sensitivity and specificity of the model. This is because the model cannot accurately detect the distribution characteristics of the data. In this study, the number of samples per patient, which is given in Annex-1, is examined on a class basis, and it is seen that the Normal state is much more numerous than the Apnea and Hypopnea conditions. On average, 4.3 M of 4.5 M records represent Normal status. Since the severe data imbalance situation is inherent in the sleep recording and scoring process, it was not possible to change it. For this reason, we under-sampling according to the "majority" class with the SMOTE method to eliminate the problem. Thus, we ensured that it was transferred to the learning stage by reducing the sample only in the most dominant class. The sample numbers after this process were again as given in Annex-1.

### Feature extraction and processing problem

In the feature extraction and processing phase, the identification and selection of the features and the selection of the appropriate classifier for this selection are laborious and require prior knowledge. As described in this study, four different types of deep learning models and 11 different machine learning algorithms were used. More importantly, unlike all other studies, sleep disorders were detected with many 23 features. While the use of many and various features enables the proposed model to achieve positive results in terms of classification performance, it is disadvantageous in terms of required processing power. For this reason, the proposed model is thought to be suitable for clinics or hospitals that want to make a more comprehensive and detailed analysis instead of personal home users. Detailed results about the patient can be obtained by detecting with high performance in six different classes. Fewer features or a single-layer classifier design means lower performance and a simpler binary classification.

### Two-tier model design

The proposed two-layer design has been used to improve the overall performance and to find solutions to some of the limitations of deep learning models. The design uses the decision-making algorithm in a second layer. In this way, it eliminates some addictions of the patient. This is quite valuable. Because sleep state is highly dependent on the age, gender, and habits of the person.

### Future work

Efficient and high-performance automated systems are needed to detect sleep disorder and help determine sleep apnea events and apply the most appropriate treatment. These systems are important so that people can be diagnosed quickly. The proposed method aims to make a high-performance and detailed analysis by processing 23 different data and making a comprehensive analysis. With the proposed method, it was preferred to use more complex and more features by preferring better detection. However, in this system, collecting 23 different sensor data can be quite troublesome. In the future, studies will be carried out on the selection and reduction of features to achieve the same performances with fewer features. In case of integration with wearable technologies, usage prevalence will be gained. With this integration, the discrete signal data collected during the night will be transferred to the computer environment thanks to the embedded software, and then the detection will be made. The system is not a real-time strategy, but it is of practical importance as a decision support system in detecting sleep disorders. More research on the problem is needed.

## CONCLUSION

In this study, a two-layer sleep disorder detection model working with 23 different sensor data is proposed. The model uses the distinctive features of different sensor data for learning and can detect both sleep disorders and apnea events. A comprehensive feature set was studied for 50 patients. With the use of meta-learner in the 2nd layer, some limitations of deep learning models have been solved and high performance has been achieved. Several

pre-processing processes and under-sampling methods are used for data imbalance. Experimental results confirm that the proposed model is successful and reliable when compared with similar models. The model, which uses a comprehensive analysis structure, is suitable for detailed sleep analysis. We believe that the proposed method is a good decision support system for clinical applications.

### Funding
The authors received no funding for this work.

### Competing Interests
The authors declare that they have no competing interests.

### Author Contributions
- Recep Sinan Arslan conceived and designed the experiments, performed the experiments, analyzed the data, performed the computation work, prepared figures and/or tables, authored or reviewed drafts of the article, and approved the final draft.

### Data Availability
The raw health data and source code samples are available in the Supplemental Files.

### Supplemental Information
Supplemental information for this article can be found online at http://dx.doi.org/10.7717/peerj-cs.1554#supplemental-information.

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
