# Peer review of "Sleep disorder and apnea events detection framework with high performance using two-tier learning model design"

_PeerJ Computer Science, doi:10.7717/peerj-cs.1554_

## Round 0.1 · original submission · Major Revisions

As per the reviewer comments, kindly revise the manuscript and resubmit.

Reviewer 1 ·

Basic reporting

- In defining improvements, there are two sentences which are "A classifier with high performance has been designed, which finds solutions to problems in learning networks thanks to its two-layer learning architecture" and "Proposed model can detect 6 different apnea types as multi-class is a candidate system to be a good decision support system with its high performance and comprehensive analysis structure in the field of sleep disorders." The author should delete these sentences because he already mentioned in another sentence the same thing.

- The sentence "All the results obtained in section 4 are explained in sections" must be checked in Grammarly. There are the same mistakes also in the rest of the manuscript.

- The sentence "For the decision-making layer, tests were carried out with 11 different machine learning models, including Logistic regression (LR), Random forest (RF), decision tree (DT), Logistic Regression (LR), Random forest (RF), Decision Tree (DT), Gaussian Naive Bayes (GNB), Linear Discriminant Analysis (LDA), Ada Boost, Gradient boosting (GB), ExtraTree (ET), Extreme Gradient Boosting (XGBoost), Support Vector Classifier (SVC), K-Nearest Neighbor(KNN) and the machine learning classifier which is used as meta-learner to guarantee to obtain best results have been employed in this work" is wrong. Please correct it. Used same ML name twice etc.

- In Figure 4, the author gave learning curves. If he wants to get smoother curves, while the epoch number is increasing, the learning rate can be reduced.

Experimental design

- In Table 1, the author presented results in different metrics like accuracy, specificity, AUC, etc. There is a specific reason for it. Because providing the results under the same metric is important for a fair comparison.

- Authors proposed a 2-layered design for the detection of sleep disorder and apnea's events separately. Instead of doing classification separately, it can be done in a single module? What is benefit doing separately?

- In Figure 1, after apnea detection, classification is continuing. It must be explained. Classification is done from the beginning just for 3 classes (obstructive, central, mixed) or not?

- 19 features belonging to the PSG records must be given in detail. What are these features?

- How is done of the separation(percentage) of test and training data? Validation is used or not?

- To recover the imbalanced data, preprocessing was done. But the data is still imbalanced. The author is using weighting between classes?

- The model design was given as keras.model.summary(). To better understand, it can be given a better form. Also, author used deep learning model but he did not give any explanation about training. The chosen batch-size, epoch number, optimizer, learning rate, etc.

Validity of the findings

- The shared source code is containing only meta-learners. Also, deep-learning models must be shared.

- The proposed method was only conducted on the collected dataset. To see the effectiveness of the proposed method, it can also be conducted on other shared datasets.

Reviewer 2 ·

Basic reporting

1) The problem statement is not clearly mentioned in the abstract
2) The literature survey seems to be just a summary. A critical analysis on the existing literature is missing.
3) What is the source of your data?
4) How did you get the features? Are they available from your dataset (OR) Did you use any methods to extract the features?
5) I couldn't find any explanation for the deep learning methodologies given in your article. What are the implementation parameters used?
6) How did you get the results in Figure 6? Are they implemented using the same dataset?
7) What is meant by sleep apnea events detection in figure 8?
8) What is the new contribution/novelty in your work?
9) How do you say that your methods are free from overfitting?

Experimental design

already given

Validity of the findings

already given

Additional comments

already given

---

## Round 0.2 · accepted · Accept

I am happy to inform you that reviewers are satisfied with the revisions made by you. Therefore, I am provisionally accepting the manuscript for publication.

Reviewer 1 ·

Basic reporting

The author completed the revision process successfully.

Experimental design

The authors present details of implementation.

Validity of the findings

The author explained the convergence figure. The chosen learning rate and parameters seems to make sense.

Additional comments

The author made the necessary changes in the manuscript. It is acceptable.

Reviewer 2 ·

Basic reporting

It is fine

Experimental design

It is fine

Validity of the findings

It is fine